# Exploiting the Relationship Between Kendall's Rank Correlation and Cosine Similarity for Attribution Protection

**Fan Wang**[1,2]     **Adams Wai-Kin Kong**[1]

[1] School of Computer Science and Engineering, Nanyang Technological University
[2] Rapid-Rich Object Search (ROSE) Lab, IGP, Nanyang Technological University
`fan005@e.ntu.edu.sg`  `adamskong@ntu.edu.sg`

## Abstract

Model attributions are important in deep neural networks as they aid practitioners in understanding the models, but recent studies reveal that attributions can be easily perturbed by adding imperceptible noise to the input. The non-differentiable Kendall's rank correlation is a key performance index for attribution protection. In this paper, we first show that the expected Kendall's rank correlation is positively correlated to cosine similarity and then indicate that the direction of attribution is the key to attribution robustness. Based on these findings, we explore the vector space of attribution to explain the shortcomings of attribution defense methods using $\ell_p$ norm and propose integrated gradient regularizer (IGR), which maximizes the cosine similarity between natural and perturbed attributions. Our analysis further exposes that IGR encourages neurons with the same activation states for natural samples and the corresponding perturbed samples. Our experiments on different models and datasets confirm our analysis on attribution protection and demonstrate a decent improvement in adversarial robustness.

## 1   Introduction

Recently, the explainable artificial intelligence (XAI) has revived since deep neural networks (DNNs) are applied to more security-sensitive tasks such as medical imaging [27], criminal justice [5] and autonomous driving [20]. As one of the XAI tools, model attributions explain and measure the relative impact of each feature on the final prediction. With more non-expert practitioners being involved, it is more important for them to understand and reliably interpret the mechanism behind the outputs. Besides, EU regulators also start to enforce *General Data Protection Regulation* for more transparent interpretations on decision making based on AI [10]. Therefore, the trustworthy attribution is becoming even more crucial.

Although numerous attribution methods have been proposed in recent studies [25, 26, 29, 35, 37], it has been pointed out that they are vulnerable to attribution attacks. Different from standard adversarial attacks [3, 9, 18, 22, 31] that focus on misleading classifiers to incorrect outputs, Ghorbani et al. [8] shows that it is possible to generate visually indistinguishable images which are significantly different on their attributions, but with the same predicted label. Dombrowski et al. [6] emphasizes on targeted attack that manipulates the attributions to any predefined target attributions while keeping the model outputs unchanged. There are also black-box attacks applied on text explanations [14]. Common adversarial defense mechanisms such as adversarial training [21] and distillation [23] are not able to tackle the attribution attacks; instead, researchers turn their focus on the attribution itself.

As the differences between natural and perturbed attributions are measured by *Kendall's rank correlation* [15], which reflects the ordinal importance among features, *i.e.*, the proportion of order

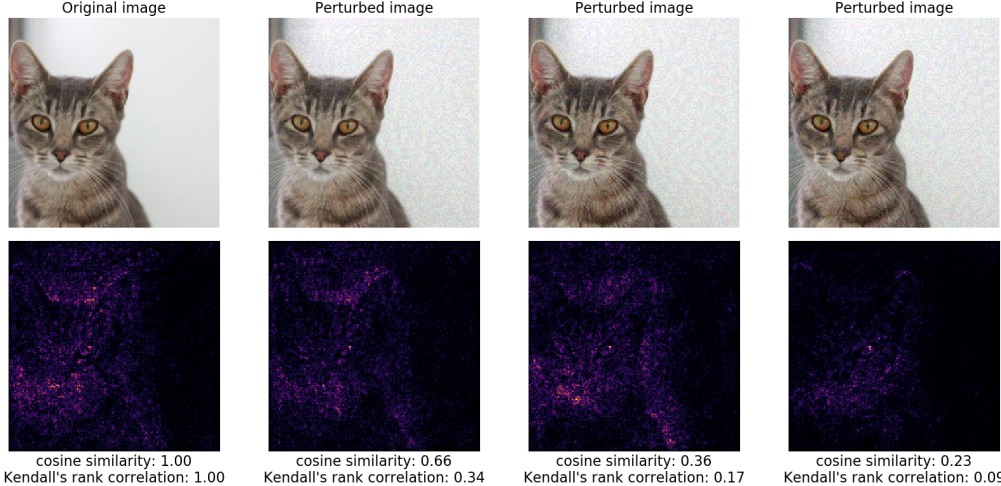

| Original image | Perturbed image | Perturbed image | Perturbed image |

| cosine similarity: 1.00 | cosine similarity: 0.66 | cosine similarity: 0.36 | cosine similarity: 0.23 |
| Kendall's rank correlation: 1.00 | Kendall's rank correlation: 0.34 | Kendall's rank correlation: 0.17 | Kendall's rank correlation: 0.09 |

Figure 1: A visualization of integrated gradients of perturbed images by restricting $\ell_1$ distance. The three perturbed attributions (the bottom images of the 2nd–4th columns) have the same $\ell_1$ distance ($d = 0.7$) to the original attribution (the bottom image of the first column). While $\ell_1$ distance remains unchanged, Kendall's rank correlations are not guaranteed to be close. However, the cosine similarities reflect the changes of the Kendall's rank coefficients.

alignment of attributions between original and perturbed images, a straightforward practice to protect the attributions against such adversaries is to maximize their Kendall's rank correlation. Since Kendall's rank correlation is not differentiable, in previous studies, it is replaced by its differentiable alternatives, such as $\ell_p$-distance regularizers [2, 4]. However, $\ell_p$-distance regularizers are not ideal for Kendall's rank correlation. As shown in Fig. 1, we found that given fixed $\ell_1$-distance between original and perturbed attributions, their Kendall's rank correlations are drastically different, which indicates $\ell_1$-distance is unstable as a measure of attribution similarity. Besides, there are also non-$\ell_p$ based regularizers, such as using Pearson's correlation, as the surrogate measurement of Kendall's rank correlation [13], it is shown to be unstable to measure the attribution.

In this paper, we discover that *cosine similarity*, as a measurement emphasizing the angle between two vectors, is consistent with Kendall's rank correlation. We present a theorem stating that cosine similarity is positively correlated with the expected Kendall's rank correlation. Based on the discovery of angular perspective, we then explain the shortcomings of $\ell_p$-norm based attribution robustness methods and propose *integrated gradients regularizer (IGR)*, an attribution robustness training regularizer that optimizes on the cosine similarity between natural and perturbed attribution. Our further analysis shows that optimizing cosine similarity encourages neurons with the same activation states. The contributions of this work are summarized as follows:

- We theoretically show that, under certain assumptions, Kendall's rank correlation between two vectors is positively correlated to their cosine similarity.

- We characterize a novel geometric perspective related to the angles between attribution vectors that explains the connection between adversarial robustness and attribution robustness for attribution methods fulfilling the axiom of completeness [29].

- Under the angular perspective, we propose *integrated gradients regularizer* (IGR) to robustly train neural networks. Our method is proved to encourage neurons with the same activation states for natural and corresponding perturbed images.

- The experimental results show that the proposed IGR regularizer can be embedded into adversarial training methods to improve their performance in terms of both attribution and adversarial robustness and outperform the state-of-the-art attribution protection methods.

The remainder of this paper is organized as follows. We first introduce the notations and previous related works in Section 2. The content starts with the theorem disclosing the relationship between Kendall's rank correlation and cosine similarity in Section 3. Based on that, we discuss the vector

space of attribution in Section 4 and describe the proposed IGR as well as its property regarding neuron activations in Section 5. Section 6 presents our experimental results and the paper concludes in Section 7.

## 2 Preliminaries and related work

Let $\{(\boldsymbol{x}^{(i)}, y^{(i)})\}_{i=1}^n$ denote data points sampled from the distribution $\mathcal{D}$, where $\boldsymbol{x}^{(i)} \in \mathbb{R}^d$ are input data and $y^{(i)} \in \{1, \ldots, k\}$ are labels. A non-bold version $x_i$ denotes the $i$-th feature of vector $\boldsymbol{x}$, and the capitalized version $X$ denotes a random variable. A classifier is the mapping from input space to the logits $f : \mathbb{R}^d \to \mathbb{R}^k$ parameterized by $\boldsymbol{\theta}$, where $f_j(\boldsymbol{x})$ is the $j$-th entry of $f(\boldsymbol{x})$, and the classification result of input $\boldsymbol{x}$ is given by the index of maximum logit $\hat{y} = \arg\max_{1 \leq j \leq k}(f_j(\boldsymbol{x}))$.

### 2.1 Attribution methods

Model attribution, denoted by $g(\boldsymbol{x})$, studies the importance that the input features contribute towards the final result. The mostly used attribution methods include perturbation-based [35, 37] and backpropagation-based methods [1], including gradient-based attribution methods [25, 26]. In particular, integrated gradients (IG) [29], one of the gradient-based methods, computes the attribution using the line integral of gradients from a baseline image $\boldsymbol{a}$ to the input image $\boldsymbol{x}$ weighted by their difference, *i.e.*,

$$g(\boldsymbol{x})_i = (x_i - a_i) \times \int_0^1 \frac{\partial f_y(\boldsymbol{a} + \alpha(\boldsymbol{x} - \boldsymbol{a}))}{\partial x_i} \, d\alpha. \tag{1}$$

IG satisfies the axiom of completeness which guarantees $\sum_i g(\boldsymbol{x})_i = f_y(\boldsymbol{x}) - f_y(\boldsymbol{a})$. We omit the baseline image $\boldsymbol{a}$ in the later parts of this paper, and it is chosen to be a black image, *i.e.*, $\boldsymbol{0}$, if not specifically stated.

### 2.2 Attribution robustness

Recent studies reveal the vulnerability of neural networks that, similar to adversarial examples, imperceptible perturbations added to natural images would have significantly different attribution while their classification results remain unchanged [8]. Heo et al. [12] manipulates the model parameters instead of input images to disturb attributions and remains high accuracy on classifications. Dombrowski et al. [6] makes targeted attack that changes original attributions to any predefined attributions and gives a theoretical explanation to this phenomenon.

Engstrom et al. [7] points out that robust optimization enhances model representations and interpretability. Chen et al. [4] and Boopathy et al. [2] use $\ell_1$-norm to constrain the distance between attributions of natural and perturbed images. Sarkar et al. [24] proposes a contrastive regularizer that emphasizes a skewed distribution on true class attribution while a uniform one on negative class attribution. Ivankay et al. [13] directly optimizes Pearson's correlation and Singh et al. [28] uses a triplet loss to minimize the upper bound of the attribution distortion. Although the previous techniques present promising results, none of them exploits the angle between attributions explicitly for attribution protection. The method introduced in this work leverages the relationship between Kendall's rank correlation and cosine similarity with a theoretical support for attribution robustness.

## 3 Kendall's rank correlation and cosine similarity

Kendall's rank correlation, often denoted by $\tau$, is a measurement of the ordinal relationship between two quantities, where two quantities have higher $\tau$ when they have more *concordant* pairs. Formally, Kendall's rank correlation between two vectors $\boldsymbol{x}$ and $\boldsymbol{x}'$ can be explicitly computed by

$$\tau = \frac{2}{d(d-1)} \sum_{i<j} \text{sign}(x_i - x_j)\text{sign}(x_i' - x_j'), \tag{2}$$

where $d$ is the dimension of the vectors. As Kendall's rank correlation is an important metric to quantify the differences between perturbed and natural attributions, we begin by presenting the relationship between Kendall's rank correlation and cosine similarity. It should be highlighted that all

the previous attribution robustness studies [2, 4, 8, 13, 28, 33] use Kendall's rank correlation as a key performance index to evaluate the effectiveness of their methods.

To enhance attribution robustness, it is equivalent to force the perturbed attribution to have a higher Kendall's rank correlation with the original one. However, as Kendall's rank correlation is not differentiable, it is difficult to directly optimize it. It is necessary to find an alternative that either approximates to or has a consistent behavior with Kendall's rank correlation. The following theorem states that cosine similarity is an appropriate replacement as a regularization term since it is positively related to the Kendall's rank correlation (Fig. 2a).

**Theorem 1.** *Given a random vector $Y = (y_1, y_2, \ldots, y_d)$ where $y_i$ follows a positive-valued distribution, and two arbitrary vectors with the same dimension, $X, X' \in \mathbb{R}^d$ that $x_i, x_i' \geq 0$, assume that there exists a sequence $\mathcal{S} = \{X_i\}_{i=1}^N$ with $X = X_0$ and $X' = X_N$, where the vectors satisfy the condition that $\cos(X_i, Y) \geq \cos(X_{i+1}, Y)$, and each $X_{i+1}$ can be induced from its previous vector $X_i$ through one of the following two operations,*

*(i) arbitrarily exchanging two entries of $X_i$*

*(ii) multiplying one entry in $X_i$ by $\alpha \in (0, 1]$*

*Then Kendall's rank correlations of $Y$ with $X$ and $X'$ have the property that $\mathbb{E}[\tau(X, Y)] \geq \mathbb{E}[\tau(X', Y)]$, where the expectation is taken over $Y$ satisfying $\cos(X_i, Y) \geq \cos(X_i + 1, Y)$.*

The full proof and discussions can be found in Appendix A. In the scenario of attribution robustness, under the above assumption, we denote $Y$ as the natural attribution $g(\boldsymbol{x})$, and $X$ and $X'$ as two perturbed attributions $g(\boldsymbol{x}')$ and $g(\boldsymbol{x}'')$. If the perturbed attribution has a greater cosine similarity with natural attribution, then their expected Kendall's rank correlation is also greater. Explicitly speaking, if $\cos(g(\boldsymbol{x}'), g(\boldsymbol{x})) \geq \cos(g(\boldsymbol{x}''), g(\boldsymbol{x}))$, then $\mathbb{E}[\tau(g(\boldsymbol{x}'), g(\boldsymbol{x}))] \geq \mathbb{E}[\tau(g(\boldsymbol{x}''), g(\boldsymbol{x}))]$. This theorem provides a theoretical foundation that supports using cosine similarity for attribution protection because it directly links to Kendall's rank correlation.

## 4 Characterization of geometric perspective on attributions

In the last section, we have indicated the relationship between cosine similarity and Kendall's rank correlation. In this section, we use this relationship to explain (i) the drawbacks of attribution protections based on $\ell_p$-norm, *e.g.*, $\min_\theta \|g(\boldsymbol{x}) - g(\tilde{\boldsymbol{x}})\|_p$ in Chen et al. [4], where $\boldsymbol{x}$ is a natural sample and $\tilde{\boldsymbol{x}}$ is a perturbed sample; (ii) the inappropriateness of standard adversarial training for attribution protection and (iii) the limitation of the cosine similarity, *i.e.*, $\min_\theta (1 - \cos(g(\boldsymbol{x}), g(\tilde{\boldsymbol{x}})))$ for standard adversarial protection. In this discussion, $g(\boldsymbol{x})$ and $g(\tilde{\boldsymbol{x}})$ are considered as vectors, and as stated in Theorem 1, a smaller angle between them implies higher attribution robustness. The attribution method $g$ is assumed to fulfill the axiom of completeness[1], *i.e.*, $f_y(\boldsymbol{x}) - f_y(\boldsymbol{a}) = \sum_i g(\boldsymbol{x})_i$. If $g(\boldsymbol{x})_i \geq 0$ for all $i$, $\|g(\boldsymbol{x})\|_2 = \sqrt{\sum_i g(\boldsymbol{x})_i^2} \leq \sum_i g(\boldsymbol{x})_i = f_y(\boldsymbol{x})$. In other words, $\|g(\boldsymbol{x})\|_2$ is the lower bound of $f_y(\boldsymbol{x})$ and larger $\|g(\boldsymbol{x})\|_2$ implies higher classification accuracy. Thus, minimizing the angle between $g(\boldsymbol{x})$ and $g(\tilde{\boldsymbol{x}})$ and maximizing their magnitudes would respectively enhance their attributional and adversarial robustness.

Fig. 2b shows a two-dimensional projection for the ease of illustration, where each 2D point represents an attribution of an input. Higher-dimensional cases can be extended in a similar manner. In Fig. 2b, $g(\boldsymbol{x})$ is the original attribution of $\boldsymbol{x}$ and the others are its perturbed counterparts. Given two attributions, $g(\boldsymbol{x}')$ and $g(\boldsymbol{x}'')$, where $\|g(\boldsymbol{x}) - g(\boldsymbol{x}')\| = \|g(\boldsymbol{x}) - g(\boldsymbol{x}'')\|$ but $\cos(g(\boldsymbol{x}), g(\boldsymbol{x}')) > \cos(g(\boldsymbol{x}), g(\boldsymbol{x}''))$, according to Theorem 1, $\tau(g(\boldsymbol{x}), g(\boldsymbol{x}'))$ is likely larger than $\tau(g(\boldsymbol{x}), g(\boldsymbol{x}''))$, implying that the attribution of $\boldsymbol{x}'$ is likely closer to the attribution of $\boldsymbol{x}$ than that of $\boldsymbol{x}''$. It explains the results in Fig. 1 and point (i), *i.e.*, drawbacks of attribution protection based on $\ell_p$-norm.

The standard adversarial training maximizes $f_y(\tilde{\boldsymbol{x}})$, where $\tilde{\boldsymbol{x}}$ is an adversarial example. In Fig. 2b, $\boldsymbol{x}'''$ has large $\|g(\boldsymbol{x}''')\|_2$, implying that $f_y(\boldsymbol{x}''')$ is also large. In other words, the classification label of $\boldsymbol{x}'''$ is well protected. However, standard adversarial training does not explicitly minimize the angle between $g(\boldsymbol{x})$ and $g(\boldsymbol{x}''')$. It implies that $\tau(g(\boldsymbol{x}), g(\boldsymbol{x}'''))$ can be small and $\boldsymbol{x}'''$ can attack the attribution successfully. It should be mentioned that adversarial training does improve attribution

---

[1]Without loss of generality, we assume $f_y(\boldsymbol{a}) = 0$ and use $\ell_2$-norm as the illustration.

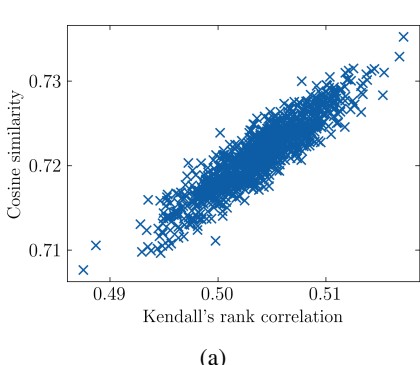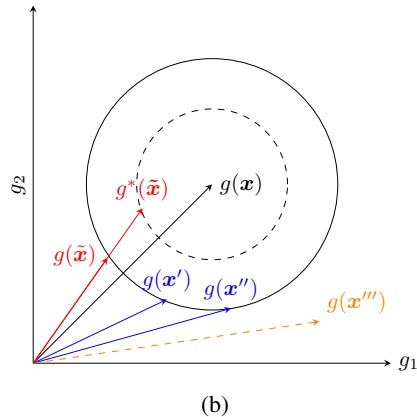

|     |     |
| :-: | :-: |
| (a) | (b) |

Figure 2: (a) Visualization of Kendall's rank correlation and cosine similarity using simulated data. Given a fixed vector $\boldsymbol{u}$ with dimension of 10,000, one thousand random vectors $\boldsymbol{v}_i$ are sampled and their corresponding $\tau$ and $\cos$ with $\boldsymbol{y}$ are calculated and plotted. The positive correlation can be observed and is later proved by Theorem 1. (b) 2D illustration of comparison of attribution trained by $\ell_p$-norm and cosine similarity. The axises are two dimensions of attribution. The solid ball and dashed ball represent two networks. Solid ball represents the untrained attribution surface $g$ and dashed ball is the trained surface $g^*$.

robustness because it smooths the decision surface [33] although it is not the most ideal one. It explains the point (ii).

Point (iii) can be explained similarly. Since the cosine similarity, or $\min_\theta (1 - \cos(g(\boldsymbol{x}), g(\tilde{\boldsymbol{x}}))$, does not necessarily enlarge the magnitude of $g(\tilde{x})$, it cannot improve network robustness against standard adversarial attack. Fig. 2b shows two networks (the dashed circle and solid circle). $\cos(g(\boldsymbol{x}), g(\tilde{\boldsymbol{x}})) = \cos(g(\boldsymbol{x}), g^*(\tilde{\boldsymbol{x}}))$, which implies that the two networks perform the same on attribution protection, but $\|g^*(\tilde{\boldsymbol{x}})\| > \|g(\tilde{\boldsymbol{x}})\|$, meaning $g^*$ is more robust against the standard adversarial attack from $\tilde{\boldsymbol{x}}$, while $g$ is more vulnerable.

To protect against both attribution attack and adversarial attack, in the following section, the proposed IGR is optimized with adversarial loss together, where the former minimizes the angle between attribution vectors to perform attribution protection, and the latter maximizes their magnitude to offer standard adversarial protection.

## 5 Integrated gradients regularizer (IGR)

Based on the above analysis, in this section, we introduce the integrated gradients regularizer (IGR), which regularizes the cosine similarity between natural and perturbed attributions.

### 5.1 IGR robust training objective

Since Kendall's rank correlation and cosine similarity are positively related, we suggest to improve attribution robustness, especially integrated gradients, by maximizing the cosine similarity between natural and perturbed attributions, or equivalently, minimizing $1 - \cos(\text{IG}(\boldsymbol{x}), \text{IG}(\tilde{\boldsymbol{x}}))$. Therefore, we propose the following training objective function incorporating the IGR

$$\mathcal{L}_{igr} = \mathbb{E}_{\mathcal{D}}[\mathcal{L}(\tilde{\boldsymbol{x}}, y; \boldsymbol{\theta}) + \lambda (1 - \cos(\text{IG}(\boldsymbol{x}), \text{IG}(\tilde{\boldsymbol{x}})))], \tag{3}$$

where $\mathcal{L}$ is a standard loss function used in robust training, and $\lambda$ is a hyper-parameter. We will later show in Section 6 that $\mathcal{L}$ can be chosen from existing loss function in robust training, and our IGR will further improve the robustness upon baseline methods.

In practice, the integral inside IG definition can be numerically computed by its Riemann sum, and IG is approximated by

$$\hat{\text{IG}}(\boldsymbol{x})_i = x_i \times \frac{1}{m} \sum_{k=1}^{m} \frac{\partial f_y(\frac{k}{m}\boldsymbol{x})}{\partial x_i}. \tag{4}$$

Similar to adversarial training, optimizing the above objective function in Eq. (3) requires perturbed example $\tilde{\boldsymbol{x}}$ that maximally diverts its IG from the original counterpart. Such examples can be found by maximizing the proposed regularizer within its $\ell_p$-ball with radius $\varepsilon$, i.e.,

$$\tilde{\boldsymbol{x}} = \underset{\tilde{\boldsymbol{x}} \in \mathcal{B}_\varepsilon(\boldsymbol{x})}{\arg\max} \left(1 - \cos(\text{IG}(\boldsymbol{x}), \text{IG}(\tilde{\boldsymbol{x}}))\right). \tag{5}$$

It is noticed that computing the adversarial loss $\mathcal{L}(\tilde{\boldsymbol{x}}, y; \boldsymbol{\theta})$ itself relies on $\tilde{\boldsymbol{x}}$, which can be obtained from adversarial attacks, i.e., $\tilde{\boldsymbol{x}} = \arg\max_{\tilde{\boldsymbol{x}}} \mathcal{L}(\tilde{\boldsymbol{x}}, y; \boldsymbol{\theta})$. Thus, here we reuse these $\tilde{\boldsymbol{x}}$ in IGR to avoid repeatedly using gradient descent methods to find the optimum in Eq. (5) and speed up the training. For example, if $\mathcal{L}$ is the standard adversarial training loss function, we directly use the examples generated from PGD attack. The computation cost of IGR is similar to previous proposed methods.

The use of Pearson's correlation regularizer in Ivankay et al. [13] as the replacement of Kendall's rank correlation is the closest method to ours. Ivankay et al. [13] suggests that Pearson's correlation regularizer keeps the ranking of feature constant. However, the statement is not supported by any theoretical justification while we give a theorem that shows cosine similarity is positively related to Kendall's rank correlation. Besides, the Pearson's correlation is an unstable metric for attributions with small variances. For a fixed vector, two slightly different inputs $\boldsymbol{\delta}$ can have drastically different Pearson's correlation, which easily fluctuate from $-1$ to $1$. The detailed discussion can be found in Appendix B.

## 5.2 IGR induces more consistent neuron activation states

An interesting discovery about IGR is related to neuron activations. We found that the activation functions in ReLU networks trained with IGR are more often with the same neuron activation states for natural sample and corresponding perturbed sample. For deep networks with ReLU activations, if the pre-ReLU value is positive (negative) for natural sample, the probability of pre-ReLU value being positive (negative) for corresponding perturbed sample would increase when trained with IGR. To analyze this phenomenon, a single-layer neural network with ReLU activation is studied. The results from this single-layer neural network can be extended to deep networks by stacking multiple layers.

Recall that $\boldsymbol{x} \in \mathbb{R}^d$ is an input image, and the network function $f$ is parameterized by $(\boldsymbol{W}, \boldsymbol{u}, c) \in \mathbb{R}^{d \times m} \times \mathbb{R}^m \times \mathbb{R}$, where $\boldsymbol{W}_i$ is the column vector of $\boldsymbol{W}$, $w_{ij}$ is the $ij$-th entry of matrix $\boldsymbol{W}$ and $u_i$ is the $i$-th entry of vector $\boldsymbol{u}$, i.e., $f(\boldsymbol{x}) = \boldsymbol{u}^\top \text{ReLU}(\boldsymbol{W}^\top \boldsymbol{x}) + c$. Then, the following proposition holds.

**Proposition 1.** *Given a single-layer neural network with ReLU activation, and with the above parameterization, if, for all $i$, $\boldsymbol{W}_i$ and $u_i$ are all independent and identically distributed random variables following Gaussian distributions, i.e., $\boldsymbol{W}_i \overset{i.i.d.}{\sim} \mathcal{N}(0, \sigma_w^2 I_d)$ and $u_i \overset{i.i.d.}{\sim} \mathcal{N}(0, \sigma_u^2)$, and two input images that each has small variance, $\boldsymbol{x}$ and $\tilde{\boldsymbol{x}}$, then*

$$\cos(IG(\boldsymbol{x}), IG(\tilde{\boldsymbol{x}})) \approx \frac{\mathbb{P}(W^\top \boldsymbol{x} > 0 \cap W^\top \tilde{\boldsymbol{x}} > 0)}{\sqrt{\mathbb{P}(W^\top \boldsymbol{x} > 0)\mathbb{P}(W^\top \tilde{\boldsymbol{x}} > 0)}}. \tag{6}$$

The proof can be found in Appendix A.2. The right-hand side of Eq. (6) is called the *activation consistency* of natural and perturbed samples. For the sake of convenience, let the event $W^\top \boldsymbol{x} > 0$ be $A$ and $W^\top \tilde{\boldsymbol{x}} > 0$ be $B$. The right-hand side of Eq. (6) can be rewritten as $\mathbb{P}(A \cap B)/\sqrt{\mathbb{P}(A)\mathbb{P}(B)}$. Since $\mathbb{P}(A \cap B)$ has the upper bound that $\mathbb{P}(A \cap B) \leq \min\{\mathbb{P}(A), \mathbb{P}(B)\}$, it is obvious that

$$\frac{\mathbb{P}(A \cap B)}{\sqrt{\mathbb{P}(A)\mathbb{P}(B)}} \leq \frac{\mathbb{P}(A \cap B)}{\sqrt{\mathbb{P}(A \cap B)\mathbb{P}(A \cap B)}} = 1, \tag{7}$$

and the equality holds when event $A$ happens with the same probability as event $B$, i.e., $\mathbb{P}(A) = \mathbb{P}(B) = \mathbb{P}(A \cap B)$; alternatively, the equality can hold when $\mathbb{P}(A^c) = \mathbb{P}(B^c) = \mathbb{P}(A^c \cup B^c)$. In other words, maximizing Eq. (6) encourages that the network activates the same set of neurons for $\boldsymbol{x}$ and $\tilde{\boldsymbol{x}}$, or deactivates the same set of neurons for $\boldsymbol{x}$ and $\tilde{\boldsymbol{x}}$.

# 6 Experiments and results

## 6.1 Experimental configurations

We evaluate the performance of IGR on different datasets, including MNIST [19], Fashion-MNIST [34] and CIFAR-10 [17]. For MNIST and Fashion-MNIST, we use a network consisting of four

Table 1: A summary of loss functions used in AT, TRADES and MART, and added with IGR

| Model | Loss function |
|---|---|
| AT ($\mathcal{L}_{at}$) | $\text{CE}(f(\tilde{\boldsymbol{x}}), y)$ |
| TRADES ($\mathcal{L}_{trades}$) | $\text{CE}(f(\tilde{\boldsymbol{x}}), y) + \beta\text{KL}(f(\boldsymbol{x})\|f(\tilde{\boldsymbol{x}}))$ |
| MART ($\mathcal{L}_{mart}$) | $\text{BCE}(f(\tilde{\boldsymbol{x}}), y)$ |
| | $+\beta\text{KL}(f(\boldsymbol{x})\|f(\tilde{\boldsymbol{x}}))(1 - f_y(\boldsymbol{x}))$ |
| +IGR | $+\lambda\left(1 - \cos(\text{IG}(\boldsymbol{x}), \text{IG}(\tilde{\boldsymbol{x}}))\right)$ |

Table 2: Attribution robustness of models trained by different defense methods under IFIA (top-k).

| Model | MNIST | | Fashion-MNIST | | CIFAR-10 | |
|---|---|---|---|---|---|---|
| | top-k | Kendall | top-k | Kendall | top-k | Kendall |
| Standard | 32.21% | 0.0955 | 42.83% | 0.1884 | 46.71% | 0.1662 |
| IG-NORM [4] | 36.13% | 0.1562 | 51.84% | 0.3446 | 74.49% | 0.5811 |
| IG-SUM-NORM [4] | 41.53% | 0.2240 | 57.27% | 0.4097 | 78.70% | 0.6901 |
| AdvAAT [13] | 51.74% | 0.3791 | 73.62% | 0.5810 | 72.11% | 0.5484 |
| ART [28] | 30.38% | 0.1439 | 31.71% | 0.2079 | 70.44% | 0.6875 |
| SSR [33] | 38.77% | 0.1650 | 60.40% | 0.4321 | 71.20% | 0.5498 |
| AT [21] | 34.35% | 0.1846 | 32.00% | 0.1516 | 72.21% | 0.5578 |
| AT+IGR | 33.40% | 0.1582 | 53.36% | 0.3750 | 73.37% | 0.5775 |
| TRADES [36] | 36.37% | 0.2127 | 57.01% | 0.2582 | 78.28% | 0.6903 |
| TRADES+AdvAAT | 52.04% | 0.4315 | 79.15% | 0.5794 | 71.30% | 0.5239 |
| TRADES+IGR | **56.13%** | **0.4537** | **80.62%** | **0.6565** | **80.26%** | **0.6940** |
| MART [32] | 32.50% | 0.1261 | 58.57% | 0.4262 | 76.11% | 0.6192 |
| MART+IGR | 37.34% | 0.1854 | 57.97% | 0.4317 | 76.56% | 0.6328 |

convolutional layers followed by three fully connected layers. The model is trained by Adam Optimizer [16] with learning rate $10^{-4}$ for 90 epochs. For CIFAR-10, we train a ResNet-18 [11] for 120 epochs using SGD [30] with initial learning rate 0.1, momentum 0.9 and weight decay $5 \times 10^{-4}$. The learning rate decays by 0.1 at the 75th and 90th epoch. All the experiments are run on NVIDIA GeForce RTX 3090.

As discussed in Section 5, IGR is applied with state-of-the-art adversarial training methods: standard adversarial training (AT)[21], TRADES [36] and MART [32]. $\mathcal{L}_{at}$, $\mathcal{L}_{trades}$ and $\mathcal{L}_{mart}$ in Table 1 are the objective functions of these methods, and are regarded as $\mathcal{L}$ in Eq. (3). In Table 1, CE denotes the cross-entropy loss and KL denotes the KL-divergence. BCE is a boosted cross-entropy (see details in Wang et al. [32]). Note that both AT and MART generate adversarial examples by maximizing the CE loss, while TRADES maximizes the KL-divergence regularizer. Following the baseline methods, we directly leverage the perturbed examples generated by their original techniques to compute the integrated gradients, as well as the IGR, instead of generating our own ones using Eq. (5). Moreover, to ensure fair comparisons, we keep the hyper-parameters the same for models with or without IGR.

## 6.2 Evaluation on attribution robustness

To evaluate our method under attribution attack, the iterative feature importance attacks (IFIA) using top-k intersection as dissimilarity function (*top-k*) [8] is adapted. IFIA generates perturbations by iteratively maximizing the dissimilarity function that measures the changes between attributions of images, while keeps the classification results unchanged. In this experiment, we perform 200-step IFIA as in Chen et al. [4]. For MNIST and Fashion-MNIST, we choose $k = 100$ and the perturbation size $\varepsilon = 0.3$. For CIFAR-10, $k = 1000$ and $\varepsilon = 8/255$. Two metrics are chosen to evaluate the performance under attribution attack as in Chen et al. [4]: top-k intersection and Kendall's rank correlation, where top-k intersection counts the proportion of pixels that coincide in the $k$ most important features. Each sample is attacked five times and the mean metrics are reported. For both metrics, a higher number indicates that the model is more robust under the attack.



| Image | Original IG (baseline) | Adv IG (baseline) | Original IG (+IGR) | Adv IG (+IGR) |

Figure 3: IGR improves attribution robustness. The second and third images are the IG of the original and perturbed image on the baseline model. The last two images are respectively the IG of the original image and perturbed image from the model trained with IGR. Both baseline and baseline+IGR models make the correct classifications, while only baseline+IGR protects the attribution of perturbed image. More visualization results are given in Appendix C

Table 3: Adversarial accuracy (%) of CNN trained by different defense methods on MNIST, Fashion-MNIST and CIFAR-10.

| | MNIST | | | | Fashion-MNIST | | | | CIFAR-10 | | | |
|---|---|---|---|---|---|---|---|---|---|---|---|---|
| Model | Natural | FGSM | PGD20 | $CW_\infty$ | Natural | FGSM | PGD20 | $CW_\infty$ | Natural | FGSM | PGD20 | $CW_\infty$ |
| AT | 99.43 | 99.39 | 99.25 | 99.24 | **89.75** | 78.92 | 74.69 | 74.65 | 73.09 | 69.42 | 37.56 | 45.28 |
| +IGR | **99.51** | **99.45** | **99.32** | **99.32** | 80.98 | **79.20** | **76.79** | **76.31** | **73.69** | **70.40** | **38.21** | **46.70** |
| TRADES | 99.40 | 99.36 | 99.21 | 99.19 | 78.82 | 77.66 | 75.94 | 75.58 | 81.33 | 79.15 | **55.02** | 52.40 |
| +IGR | 99.40 | **99.40** | **99.26** | **99.24** | **80.61** | **79.05** | **76.89** | **76.44** | **81.65** | **79.54** | 54.65 | **52.43** |
| MART | 99.39 | 99.29 | 99.09 | 99.08 | 79.43 | 79.36 | 77.91 | 77.49 | 78.97 | 77.19 | 56.05 | 50.99 |
| +IGR | **99.51** | **99.39** | **99.28** | **99.24** | **81.51** | **82.13** | **79.93** | **79.01** | **79.27** | **77.35** | **56.47** | **51.11** |

To compare, attribution protection methods, IG-NORM and IG-SUM-NORM by Chen et al. [4], *Smooth Surface Regularization (SSR)* [33], *Attributional Robustness Training (ART)* [28] and *Adversarial Attributional Training* with robust training loss (*AdvAAT*), are implemented and evaluated on all the datasets. A cross-entropy loss trained natural model (*standard*) is also included as a baseline. The details of these baseline methods are briefly introduced in Appendix C.2.

From the results in Table 2, we observed the following phenomenons. (i) Compared with baseline methods (AT, TRADES and MART), models trained with IGR outperform their corresponding counterparts in terms of both top-k intersection and Kendall's rank correlation. (ii) Adversarial defense methods themselves also help the attribution protection, especially improve on Fashion-MNIST and CIFAR-10 comparing with the standard cross-entropy training. (iii) Compared with other attribution protection methods, standard adversarial defense methods, including AT, TRADES and MART, are weaker in attribution robustness; however, they achieve comparable or even stronger attribution protections when training with IGR. (iv) TRADES itself has the best attribution protections among models without IGR, and IGR provides the most significant boost on TRADES. (v) Since AdvAAT uses Pearson's correlation as a regularizer, which is close to IGR, and TRADES+IGR outperforms the other baselines, we apply Pearson's correlation on TRADES, *i.e.*, TRADES +AdvAAT, in the Table 2 for the comparison. Table 2 shows clearly that TRADES+AdvAAT does not always improve over TRADES and is consistently outperformed by TRADES+IGR.

A visualization of attribution robustness is also presented in Figure 3. It is observed that the attribution of the baseline model is easily corrupted. For model trained with IGR, although the magnitudes of IG are different from IG of the original images, the directions remain nearly identical, which is also aligned with human visual perceptions.

### 6.3    Evaluation on white-box adversarial robustness

To evaluate the performance of IGR on adversarial robustness, the trained defense models are evaluated under white-box adversarial attacks, including FGSM [9], PGD [21] and $CW_\infty$ [3], where all the attacks have the information of the entire models, including architectures and parameters. The numbers are reported in both natural accuracy (Natural) and adversarial accuracies under FGSM, PGD with 20 steps (PGD20), and $CW_\infty$ attacks. The maximum allowable perturbations are chosen

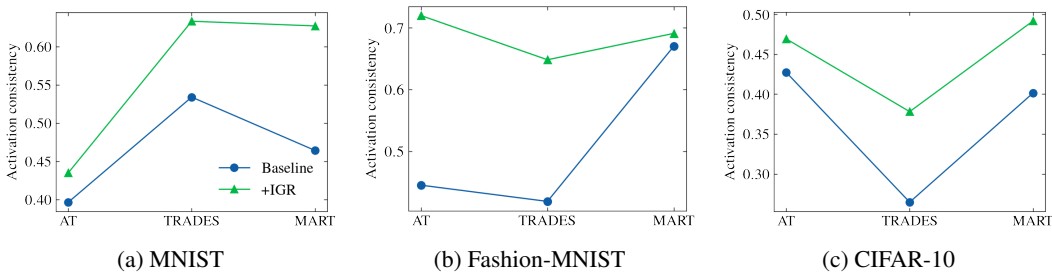

Figure 4: Activation consistency of baseline models and IGR models in MNIST, Fashion-MNIST and CIFAR-10.

to be $\varepsilon = 0.3$ for MNIST and Fashion-MNIST, and $\varepsilon = 8/255$ for CIFAR-10 as previous studies. The white-box adversarial accuracy results, as well as natural accuracy are reported in Table 3.

As shown in Table 3, defense methods trained with IGR achieve higher accuracies under all three types of attacks upon their corresponding baseline methods, except TRADES+IGR under PGD attack in CIFAR-10. In the meantime, classification accuracies of natural images are also improved in seven out of nine evaluations. This suggests that training with IGR improves adversarial accuracies without losing the generalization of natural accuracies. Although IGR is designed for attribution protection, these improvements is considered as a side-effect and a rigorous study of the phenomenon can be future work.

### 6.4  Evaluation on activation consistency

This section reports the experimental results that verify the claim in Section 5.2 — IGR encourages that the network activates the same set of neurons for natural and perturbed samples $x$ and $\tilde{x}$. During the experiments, all the pre-activation values are recorded and used to compute the proportion of nonnegative values. Thus, the activation consistency defined on the right-hand side of Eq. (6) can be numerically computed.

Fig. 4 compares the activation consistency on the baselines and the corresponding models trained with IGR. It is noticed that for all the datasets, the activation consistency of the models trained with IGR are consistently greater than the corresponding baseline models, which verifies our theory in Section 5.2. Moreover, as reported in Table 2, the improvements of AT+IGR in MNIST and MART+IGR in Fashion-MNIST are not as significant as others. The results are also reflected on activation consistency, as the value of activation consistency slightly improves from 0.40 to 0.43 on AT+IGR in MNIST and from 0.67 to 0.69 on MART+IGR in Fashion-MNIST, while TRADES+IGR that boosts the most in attribution robustness also increases the most in activation consistency.

## 7  Conclusions

In order to leverage the non-differentiable Kendall's rank correlation for attribution protection, this paper starts with a theorem indicating the positive correlation between cosine similarity and Kendall's rank correlation. We then introduce a geometric perspective to explain the shortcomings of $\ell_p$ based attribution defense methods and propose the integrated gradients regularizer to improve attribution robustness. It is discovered that IGR encourages networks activating the same set of neurons for natural and perturbed samples. Finally, experiments show that IGR can be combined with adversarial objective functions, which simultaneously minimizes the angle between attribution vectors for attribution robustness and maximizes their magnitude to offer standard adversarial protection.

## Acknowledgments and Disclosure of Funding

This work is partially supported by the Ministry of Education, Singapore through Academic Research Fund Tier 1, RG73/21

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
