# A Proofs

## A.1 Proof and discussion of Theorem 1

**Theorem 1.** *Given a random vector $Y = (y_1, y_2, \ldots, y_d)$ where $y_i$ follows a positive-valued distribution, and two arbitrary vectors with the same dimension, $X, X' \in \mathbb{R}^d$ that $x_i, x'_i \geq 0$, assume that there exists a sequence $\mathcal{S} = \{X_i\}_{i=1}^N$ with $X = X_0$ and $X' = X_N$, where the vectors satisfy the condition that $\cos(X_i, Y) \geq \cos(X_{i+1}, Y)$, and each $X_{i+1}$ can be induced from its previous vector $X_i$ through one of the following two operations,*

*(i) arbitrarily exchanging two entries of $X_i$*

*(ii) multiplying one entry in $X_i$ by $\alpha \in (0, 1]$*

*Then Kendall's rank correlations of $Y$ with $X$ and $X'$ have the property that $\mathbb{E}\left[\tau(X, Y)\right] \geq \mathbb{E}\left[\tau(X', Y)\right]$, where the expectation is taken over $Y$ satisfying $\cos(X_i, Y) \geq \cos(X_i + 1, Y)$.*

*Proof.* To prove this theorem, we show that the property holds when $N = 2$, *i.e.*, $\mathcal{S} = \{X, X'\}$, which indicates that each one of the above operations on $X$ would preserve the order of Kendall's rank correlation. The case when $N \geq 3$ can be trivially generalized using mathematical induction.

Since $X$ and $X'$ are two arbitrary vectors, it is safe to fix $X$ that $X = (x_1, x_2, \ldots, x_d)$. To analyze the cosine similarities and Kendall's rank correlation, $X$ can be assumed to be in descending order, *i.e.*, $x_1 > x_2 > \cdots > x_d$, since the order of $X'$ and $Y$ can be changed correspondingly without affecting the cosine similarities and Kendall's rank correlation. Formally, we show in the following proof that for a random vector $Y$ following exponential distribution and an arbitrary vector $X$, if the cosine similarities satisfy that $\cos(X, Y) \geq \cos(X', Y)$, then their corresponding Kendall's rank correlations have the property that $\mathbb{E}\left[\tau(X, Y)\right] \geq \mathbb{E}\left[\tau(X', Y)\right]$, where $X'$ is generated from $X$ by (1) exchanging two entries and (2) scalar multiplications.

**(1) Preservation under exchanging** Following the assumption, we consider a random vector $Y = (y_1, y_2, \ldots, y_d)$ where $y_i$ positively distributed, and another vector $X = (x_1, x_2, \ldots, x_d)$. We define the new vector $X'$ that is produced by arbitrarily exchanging two entries in $X$. Suppose we exchange the $p$-th and $q$-th entry in $X$, where $1 \leq p < q \leq d$, then $X' = (x_1, \ldots, x_{p-1}, x_q, x_{p+1}, \ldots, x_{q-1}, x_p, x_{q+1}, \ldots, x_d)$. We further assume both $X$ and $Y$ are normalized, *i.e.*, $\|X\| = \|Y\| = \|X'\| = 1$.

Now if we consider the cosine similarity and the assumption that $\cos(X, Y) > \cos(X', Y)$, we then have

$$\cos(X, Y) = \sum_{i=1}^d x_i y_i > \cos(X', Y) = \sum_{i \neq p, q}^d x_i y_i + x_p y_q + x_q y_p, \tag{8}$$

which can be simplified as

$$(x_p - x_q)(y_p - y_q) > 0 \tag{9}$$

For Kendall's rank correlation, we denote that $\Omega(X, Y) = \sum_{i<j} \text{sign}(x_i - x_j)\text{sign}(y_i - y_j)$, and $\tau(X, Y) = \frac{2}{d(d-1)}\Omega(X, Y)$. We notice that the difference between $\Omega(X, Y)$ and $\Omega(X', Y)$ only occurs when $x_p$ and $x_q$ are involved. We can write down the explicit expression of $\Omega(X, Y) - \Omega(X', Y)$

$$\begin{aligned}
\Omega(X, Y) - \Omega(X', Y) = & \left(\text{sign}(x_p - x_q) - \text{sign}(x_q - x_p)\right)\text{sign}(y_p - y_q) \\
& + \sum_{p<i<q} \left(\text{sign}(x_p - x_i) - \text{sign}(x_q - x_i)\right)\text{sign}(y_p - y_i) \\
& + \sum_{p<i<q} \left(\text{sign}(x_i - x_q) - \text{sign}(x_i - x_p)\right)\text{sign}(y_i - y_q) \tag{10} \\
= & 2 + 2\sum_{p<i<q} \left(\text{sign}(y_p - y_i) + \text{sign}(y_i - y_q)\right) \tag{11}
\end{aligned}$$

Since

$$\mathbb{E}\left[\sum_{p<i<q}\text{sign}(y_p-y_i)+\text{sign}(y_i-y_q)\,\middle|\,y_p-y_q>0\right]\geq 0, \tag{12}$$

we then have,

$$\mathbb{E}\left[\Omega(X,Y)\right]\geq\mathbb{E}\left[\Omega(X',Y)\right]. \tag{13}$$

**(2) Preservation under scalar multiplication**    We use the assumptions mentioned before that $y_i$ is positive-valued, and $x_1>x_2>\cdots>x_d>0$. Without loss of generality, we multiply $x_1$ by a scalar $\alpha\in[0,1]$, such that $x_2>\alpha x_1>x_3$, *i.e.*, $X'=(\alpha x_1,x_2,\ldots,x_d)$.

To compare $\tau(X,Y)$ and $\tau(X',Y)$, it is noticed that, under our assumptions, only the sign of $y_1-y_2$ is needed as other terms in $\Omega$ are equal for $\Omega(X,Y)$ and $\Omega(X',Y)$,

$$\Omega(X,Y)-\Omega(X',Y)=\text{sign}(x_1-x_2)\text{sign}(y_1-y_2)-\text{sign}(\alpha x_1-x_2)\text{sign}(y_1-y_2)=2\text{sign}(y_1-y_2). \tag{14}$$

Under the condition that the cosine similarity $\cos(X,Y)>\cos(X',Y)$, we have

$$x_1y_1+x_2y_2+\cdots+x_dy_d\geq\frac{\alpha x_1y_1+x_2y_2+\cdots+x_dy_d}{\sqrt{\alpha^2x_1^2+x_2^2+\cdots+x_d^2}}. \tag{15}$$

Note that $\|X\|=\|Y\|=1$. For simplicity, we denote that $A=\sqrt{\alpha^2x_1^2+x_2^2+\cdots+x_d^2}$, and it is obvious that $\alpha\leq A\leq 1$, where $A$ is close to 1 when $d$ is large,

$$x_1y_1+x_2y_2+\left(1-\frac{1}{A}\right)\left(\sum_{i=3}^{d}x_iy_i\right)\geq\frac{\alpha x_1y_1+x_2y_2}{A}, \tag{16}$$

which can be relaxed as

$$x_1y_1+x_2y_2\geq\frac{\alpha x_1y_1+x_2y_2}{A}, \tag{17}$$

*i.e.*,

$$y_1\geq Ky_2 \tag{18}$$

where $K=\frac{(1-A)x_2}{(A-\alpha)x_1}>0$. Thus, we want to show that

$$\mathbb{E}\left[\text{sign}(y_1-y_2)\,\middle|\,y_1\geq Ky_2\right]=\mathbb{P}(\text{sign}(y_1-y_2)\geq 0|y_1\geq Ky_2)-\mathbb{P}(\text{sign}(y_1-y_2)<0|y_1\geq Ky_2) \tag{19}$$

$$=2\mathbb{P}(\text{sign}(y_1-y_2)\geq 0|y_1\geq Ky_2)-1\geq 0 \tag{20}$$

We consider two cases when $K\geq 1$ and $0<K<1$. In the case that $K\geq 1$, it is obvious that

$$\mathbb{P}(\text{sign}(y_1-y_2)\geq 0|y_1\geq Ky_2)=1. \tag{21}$$

If $0<K<1$,

$$\mathbb{P}(\text{sign}(y_1-y_2)\geq 0|y_1\geq Ky_2)=\frac{\mathbb{P}(\text{sign}(y_1-y_2)\geq 0\cap y_1\geq Ky_2)}{\mathbb{P}(y_1\geq Ky_2)} \tag{22}$$

$$=\frac{\mathbb{P}(\text{sign}(y_1-y_2)\geq 0)}{\mathbb{P}(y_1\geq Ky_2)}\geq\frac{1}{2} \tag{23}$$

Thus, after combining the above two cases, we have

$$\mathbb{P}(\text{sign}(y_1-y_2)\geq 0|y_1\geq Ky_2)\geq\frac{1}{2} \tag{24}$$

which concludes our proof.

$\square$

**Discussions** In practice, the attribution values are taken absolute values to emphasize the importance of features, regardless of whether the impact are positive or negative. Thus, without loss of generality, $y_i$ is assumed to follow a positive-valued distribution in Theorem 1. We also consider the existence of the sequence $\mathcal{S}$ as an assumption that assist the formulation of the theorem. Although searching for such sequence of every pair of attributions $X$ and $X'$ can be a combinatorial problem and is constrained by computation power, the numerical simulations of finding such sequences in lower dimensions still show a high success rate ($\geq 0.8$ when $d \leq 10$), and the number of possible sequences increases drastically when the dimension is higher.

## A.2 Proof of Proposition 1

**Proposition 1.** *Given a single-layer neural network with ReLU activation, and with the above parameterization, if, for all $i$, $\boldsymbol{W}_i$ and $u_i$ are all independent and identically distributed random variables following Gaussian distributions, i.e., $\boldsymbol{W}_i \overset{i.i.d.}{\sim} \mathcal{N}(0, \sigma_w^2 I_d)$ and $u_i \overset{i.i.d.}{\sim} \mathcal{N}(0, \sigma_u^2)$, and two input images that each has small variance, $\boldsymbol{x}$ and $\tilde{\boldsymbol{x}}$, then*

$$\cos(IG(\boldsymbol{x}), IG(\tilde{\boldsymbol{x}})) \approx \frac{\mathbb{P}(W^\top \boldsymbol{x} > 0 \cap W^\top \tilde{\boldsymbol{x}} > 0)}{\sqrt{\mathbb{P}(W^\top \boldsymbol{x} > 0)\mathbb{P}(W^\top \tilde{\boldsymbol{x}} > 0)}}. \tag{6}$$

*Proof.* Recall that $\boldsymbol{x} \in \mathbb{R}^d$ is an input image, and the network function $f$ is parameterized by $(\boldsymbol{W}, \boldsymbol{u}, c) \in \mathbb{R}^{d \times m} \times \mathbb{R}^m \times \mathbb{R}$, where $\boldsymbol{W}_i$ is the column vector of $\boldsymbol{W}$, $w_{ij}$ is the $ij$-th entry of matrix $\boldsymbol{W}$ and $u_i$ is the $i$-th entry of vector $\boldsymbol{u}$, i.e., $f(\boldsymbol{x}) = \boldsymbol{u}^\top \text{ReLU}(\boldsymbol{W}^\top \boldsymbol{x}) + c$.

Following the above notations, we first write the function as

$$f(\boldsymbol{x}) = \boldsymbol{u}^\top \text{ReLU}(\boldsymbol{W}^\top \boldsymbol{x}) + c = \sum_{i=1}^m u_i(\boldsymbol{W}_i^\top \boldsymbol{x})\mathbb{1}_{\boldsymbol{W}_i^\top \boldsymbol{x} > 0} + c, \tag{25}$$

where $\mathbb{1}_{\{\cdot\}}$ denotes the indicator function, and its gradient

$$\nabla_{x_k} f(\boldsymbol{x}) = (\nabla f(\boldsymbol{x}))_k = \sum_{i=1}^m u_i w_{ki} \mathbb{1}_{\boldsymbol{W}_i^\top \boldsymbol{x} > 0}$$

We assume the bias terms are zeros without loss of generality, i.e., $c = 0$, and approximate the cosine similarity of IG using the small variance assumption that $\frac{1}{n}\sum_i x_i^2 - (\frac{1}{n}\sum_i x_i)^2 \approx 0$ as

$\cos(IG(\boldsymbol{x}), IG(\tilde{\boldsymbol{x}}))$

$$\approx \frac{\displaystyle\sum_{i=1}^m \sum_{j=1}^m \langle \boldsymbol{W}_i, \boldsymbol{W}_j \rangle u_i u_j \int_0^1 \mathbb{1}_{\boldsymbol{W}_i^\top (r\boldsymbol{x}) > 0}\, dr \int_0^1 \mathbb{1}_{\boldsymbol{W}_j^\top (r\tilde{\boldsymbol{x}}) > 0}\, dr}{\sqrt{\displaystyle\sum_{i=1}^m \sum_{j=1}^m \langle \boldsymbol{W}_i, \boldsymbol{W}_j \rangle u_i u_j \int_0^1 \mathbb{1}_{\boldsymbol{W}_i^\top (r\boldsymbol{x}) > 0}\, dr}\sqrt{\displaystyle\sum_{i=1}^m \sum_{j=1}^m \langle \boldsymbol{W}_i, \boldsymbol{W}_j \rangle u_i u_j \int_0^1 \mathbb{1}_{\boldsymbol{W}_i^\top (r\tilde{\boldsymbol{x}}) > 0}\, dr}} \tag{26}$$

Since $\langle \boldsymbol{W}_i, \boldsymbol{W}_j \rangle$ is close to 0 in high dimensional space when $i \neq j$, we approximate the above expression as

$$\frac{\displaystyle\sum_{i=1}^m \|\boldsymbol{W}_i\|_2^2 u_i^2 \int_0^1 \mathbb{1}_{\boldsymbol{W}_i^\top (r\boldsymbol{x}) > 0}\, dr \int_0^1 \mathbb{1}_{\boldsymbol{W}_i^\top (r\tilde{\boldsymbol{x}}) > 0}\, dr}{\sqrt{\displaystyle\sum_{i=1}^m \|\boldsymbol{W}_i\|_2^2 u_i^2 \int_0^1 \mathbb{1}_{\boldsymbol{W}_i^\top (r\boldsymbol{x}) > 0}\, dr}\sqrt{\displaystyle\sum_{i=1}^m \|\boldsymbol{W}_i\|_2^2 u_i^2 \int_0^1 \mathbb{1}_{\boldsymbol{W}_i^\top (r\tilde{\boldsymbol{x}}) > 0}\, dr}} \tag{27}$$

Notice that the indicator function is integrated from 0 to 1, which does not affect the sign of $\boldsymbol{W}_i^\top(r\boldsymbol{x})$ and $\boldsymbol{W}_i^\top(r\tilde{\boldsymbol{x}})$, i.e., the activation states. This implies that the activation states is the same for all

samples from baseline to the corresponding image. Thus, we can write the cosine similarity as

$$\frac{\sum_{i=1}^{m} \|\boldsymbol{W}_i\|_2^2 u_i^2 \mathbb{1}_{\boldsymbol{W}_i^\top \boldsymbol{x}>0} \mathbb{1}_{\boldsymbol{W}_i^\top \tilde{\boldsymbol{x}}>0}}{\sqrt{\sum_{i=1}^{m} \|\boldsymbol{W}_i\|_2^2 u_i^2 \mathbb{1}_{\boldsymbol{W}_i^\top \boldsymbol{x}>0}} \sqrt{\sum_{i=1}^{m} \|\boldsymbol{W}_i\|_2^2 u_i^2 \mathbb{1}_{\boldsymbol{W}_i^\top \tilde{\boldsymbol{x}}>0}}} \tag{28}$$

Since $\boldsymbol{W}_i$ and $u_i$ are independent random variables following Gaussian distributions, *i.e.*, $\boldsymbol{W}_i \sim \mathcal{N}(0, \sigma_w^2 I_d)$ and $u_i \sim \mathcal{N}(0, \sigma_u^2)$, when $m$ is sufficiently large, we have

$$\frac{1}{m} \sum_{i=1}^{m} \|\boldsymbol{W}_i\|_2^2 u_i^2 \mathbb{1}_{\boldsymbol{W}_i^\top \boldsymbol{x}>0} \mathbb{1}_{\boldsymbol{W}_i^\top \tilde{\boldsymbol{x}}>0} = \mathbb{E}_{W,u} \left[ \|W\|_2^2 u^2 \mathbb{1}_{W^\top \boldsymbol{x}>0} \mathbb{1}_{W^\top \tilde{\boldsymbol{x}}>0} \right] \tag{29}$$

The cosine similarity is then transformed into expectations

$$\cos(\text{IG}(\boldsymbol{x}), \text{IG}(\tilde{\boldsymbol{x}})) \approx \frac{\mathbb{E}_{W,u} \left[ \|W\|_2^2 u^2 \mathbb{1}_{W^\top \boldsymbol{x}>0} \mathbb{1}_{W^\top \tilde{\boldsymbol{x}}>0} \right]}{\sqrt{\mathbb{E}_{W,u} \left[ \|W\|_2^2 u^2 \mathbb{1}_{W^\top \boldsymbol{x}>0} \right]} \sqrt{\mathbb{E}_{W,u} \left[ \|W\|_2^2 u^2 \mathbb{1}_{W^\top \tilde{\boldsymbol{x}}>0} \right]}} \tag{30}$$

Based on the assumption on Gaussian distribution, we have $\mathbb{E}\left[\|W\|_2^2\right] = \text{tr}(\sigma_w^2 I_d) = d\sigma_w^2$ and $\mathbb{E}\left[u^2\right] = \sigma_u^2$, and

$$\cos(\text{IG}(\boldsymbol{x}), \text{IG}(\tilde{\boldsymbol{x}})) \approx \frac{\sigma_u^2 \mathbb{E}_W \left[ \|W\|_2^2 \mathbb{1}_{W^\top \boldsymbol{x}>0} \mathbb{1}_{W^\top \tilde{\boldsymbol{x}}>0} \right]}{\sigma_u \sqrt{\mathbb{E}_W \left[ \|W\|_2^2 \mathbb{1}_{W^\top \boldsymbol{x}>0} \right]} \sigma_u \sqrt{\mathbb{E}_W \left[ \|W\|_2^2 \mathbb{1}_{W^\top \tilde{\boldsymbol{x}}>0} \right]}} \tag{31}$$

$$= \frac{\mathbb{E}_W \left[ \|W\|_2^2 \mathbb{1}_{W^\top \boldsymbol{x}>0} \mathbb{1}_{W^\top \tilde{\boldsymbol{x}}>0} \right]}{\sqrt{\mathbb{E}_W \left[ \|W\|_2^2 \mathbb{1}_{W^\top \boldsymbol{x}>0} \right]} \sqrt{\mathbb{E}_W \left[ \|W\|_2^2 \mathbb{1}_{W^\top \tilde{\boldsymbol{x}}>0} \right]}} \tag{32}$$

$$= \frac{d\sigma_w^2 \mathbb{P}(W^\top \boldsymbol{x} > 0 \cap W^\top \tilde{\boldsymbol{x}} > 0)}{d\sigma_w^2 \sqrt{\mathbb{P}(W^\top \boldsymbol{x} > 0)\mathbb{P}(W^\top \tilde{\boldsymbol{x}} > 0)}} \tag{33}$$

$$= \frac{\mathbb{P}(W^\top \boldsymbol{x} > 0 \cap W^\top \tilde{\boldsymbol{x}} > 0)}{\sqrt{\mathbb{P}(W^\top \boldsymbol{x} > 0)\mathbb{P}(W^\top \tilde{\boldsymbol{x}} > 0)}} \tag{34}$$

$\square$

## B    Unstable Pearson's correlation

In this section, we discuss the unstable Pearson's correlation for small variance inputs, *i.e.*, $\frac{1}{n}\sum_{i=1}^{n} x_i^2 - \left(\frac{1}{n}\sum_{i=1}^{n} x_i\right)^2 \approx 0$. Consider the Pearson's correlation between $\boldsymbol{x}$ and $\boldsymbol{x} + \boldsymbol{\eta}$, where $\boldsymbol{\eta}$ is vector and bounded by $\|\boldsymbol{\eta}\| \leq \epsilon$ for small $\epsilon$. Then the Pearson's correlation between $\boldsymbol{x}$ and $\boldsymbol{x} + \boldsymbol{\eta}$ can be written as

$$\rho(\boldsymbol{x}, \boldsymbol{x} + \boldsymbol{\eta}) = \frac{\frac{1}{n}\sum_{i=1}^{n} x_i(x_i + \eta_i) - \left(\frac{1}{n}\sum_{i=1}^{n} x_i\right)\left(\frac{1}{n}\sum_{i=1}^{n} x_i + \eta_i\right)}{\sqrt{\frac{1}{n}\sum_{i=1}^{n} x_i^2 - \left(\frac{1}{n}\sum_{i=1}^{n} x_i\right)^2} \sqrt{\frac{1}{n}\sum_{i=1}^{n} (x_i + \eta_i)^2 - \left(\frac{1}{n}\sum_{i=1}^{n} (x_i + \eta_i)\right)^2}} \tag{35}$$

Consider the numerator of $\rho(\boldsymbol{x}, \boldsymbol{x} + \boldsymbol{\eta})$ as

$$N_{\rho(\boldsymbol{x}, \boldsymbol{x}+\boldsymbol{\eta})} = \frac{1}{n}\sum_{i=1}^{n} x_i^2 + \frac{1}{n}\sum_{i=1}^{n} x_i \eta_i - \left(\frac{1}{n}\sum_{i=1}^{n} x_i\right)^2 - \left(\frac{1}{n}\sum_{i=1}^{n} x_i\right)\left(\frac{1}{n}\sum_{i=1}^{n} \eta_i\right) \tag{36}$$

$$\approx \frac{1}{n}\sum_{i=1}^{n} x_i \eta_i - \left(\frac{1}{n}\sum_{i=1}^{n} x_i\right)\left(\frac{1}{n}\sum_{i=1}^{n} \eta_i\right) \tag{37}$$

Similarly, we can obtain

$$N_{\rho(\boldsymbol{x}, \boldsymbol{x}-\boldsymbol{\eta})} \approx -\frac{1}{n}\sum_{i=1}^{n} x_i \eta_i + \left(\frac{1}{n}\sum_{i=1}^{n} x_i\right)\left(\frac{1}{n}\sum_{i=1}^{n} \eta_i\right) \tag{38}$$

---

**Algorithm 1** Adversarial Training with IGR

---

**Input:** classifier $f$, data $\left\{\boldsymbol{x}^{(i)}, y^{(i)}\right\}_{i=1}^{n}$, number of PGD attack $n$, PGD step-size $\alpha$, maximum allowable perturbation $\varepsilon$, scaling parameter of IGR $\lambda$
**for** $epoch \in \{1, 2, \ldots\}$ **do**
    Compute IG($\boldsymbol{x}$)
    Randomly initiate $\tilde{\boldsymbol{x}} = \boldsymbol{x} + \mathcal{U}[-\varepsilon, \varepsilon]$
    **for** $i = 1$ **to** $n$ **do**
        $\tilde{\boldsymbol{x}} = \tilde{\boldsymbol{x}} + \alpha * \text{sign}(\nabla \mathcal{L}_{at}(\tilde{\boldsymbol{x}}, y))$
        $\tilde{\boldsymbol{x}} = Proj_{\mathcal{B}_\varepsilon}(\tilde{\boldsymbol{x}})$
    **end for**
    Compute IG($\tilde{\boldsymbol{x}}$)
    Compute loss $\mathcal{L}_{igr} = \mathcal{L}_{at}(\tilde{\boldsymbol{x}}, y) + \lambda(1 - \cos(\text{IG}(\boldsymbol{x}), \text{IG}(\tilde{\boldsymbol{x}})))$
    Update model parameters $\boldsymbol{\theta}$ using $\mathcal{L}_{igr}$
**end for**
**Return** $f$

---

Thus, $N_{\rho(\boldsymbol{x}, \boldsymbol{x}+\boldsymbol{\eta})} \approx -N_{\rho(\boldsymbol{x}, \boldsymbol{x}-\boldsymbol{\eta})}$. Since $\frac{1}{n}\sum_{i=1}^{n} x_i^2 - \left(\frac{1}{n}\sum_{i=1}^{n} x_i\right)^2 \approx 0$, the denominator of $\rho(\boldsymbol{x}, \boldsymbol{x}+\boldsymbol{\eta})$ and $\rho(\boldsymbol{x}, \boldsymbol{x}-\boldsymbol{\eta})$ are both small. Thus, $\rho(\boldsymbol{x}, \boldsymbol{x}+\boldsymbol{\eta})$ and $\rho(\boldsymbol{x}, \boldsymbol{x}-\boldsymbol{\eta})$ would be drastically different. However, since $\|\boldsymbol{\eta}\|$ is very small, both $\boldsymbol{x} + \boldsymbol{\eta}$ and $\boldsymbol{x} - \boldsymbol{\eta}$ are in fact close to $\boldsymbol{x}$. Therefore, the Pearson' correlation can be unstable.

## C   Additional experimental details and results

### C.1   Pseudo-code of IGR training

Algorithm 1 shows the pseudo-code for AT+IGR, where $\tilde{\boldsymbol{x}}$ is generated from PGD in adversarial training. Similarly, for TRADES+IGR and MART+IGR, $\tilde{\boldsymbol{x}}$ is obtained by replacing $\mathcal{L}_{at}$ using $\mathcal{L}_{trades}$ and $\mathcal{L}_{mart}$.

### C.2   Implementation details of baseline attribution robustness methods

The objective functions of the baseline attribution robustness methods are defined as follows.

**IG-NORM [4]**

$$\mathbb{E}_{\mathcal{D}}\left[\mathcal{L}(\boldsymbol{x}, y; \boldsymbol{\theta}) + \lambda \max_{\tilde{\boldsymbol{x}} \in \mathcal{B}_\varepsilon(\boldsymbol{x})} \|\text{IG}(\boldsymbol{x}, \tilde{\boldsymbol{x}})\|_1\right] \tag{39}$$

**IG-SUM-NORM [4]**

$$\mathbb{E}_{\mathcal{D}}\left[\max_{\tilde{\boldsymbol{x}} \in \mathcal{B}_\varepsilon(\boldsymbol{x})}\{\mathcal{L}(\tilde{\boldsymbol{x}}, y; \boldsymbol{\theta}) + \lambda\|\text{IG}(\boldsymbol{x}, \tilde{\boldsymbol{x}})\|_1\}\right] \tag{40}$$

**AdvAAT [13]**

$$\mathbb{E}_{\mathcal{D}}\left[\max_{\tilde{\boldsymbol{x}} \in \mathcal{B}_\varepsilon(\boldsymbol{x})}\{\mathcal{L}(\tilde{\boldsymbol{x}}, y; \boldsymbol{\theta}) + \lambda\text{PCL}(\text{IG}(\boldsymbol{x}), \text{IG}(\tilde{\boldsymbol{x}}))\}\right], \tag{41}$$

where $\text{PCL}(\cdot) = 1 - [\text{PCC}(\cdot) + 1]/2$ is derived from *Pearson's Correlation Coefficient* ($\text{PCC}(\cdot)$). Different from AT[21], AdvAAT adds a regularizer monitoring the attributions to the maximization problem. It generates perturbed samples that maximize both cross entropy and regularizer.

**ART [28]**

$$\mathbb{E}_{\mathcal{D}}\left[\mathcal{L}(\tilde{\boldsymbol{x}}, y; \boldsymbol{\theta}) + \lambda \log\left(1 + \exp(-(d(g^*(\boldsymbol{x}), \boldsymbol{x}) - d(g^y(\boldsymbol{x}), x)))\right)\right], \tag{42}$$

where

$$d(g^i(\boldsymbol{x}), \boldsymbol{x}) = 1 - \frac{g^i(\boldsymbol{x})^\top \boldsymbol{x}}{\|g^i(\boldsymbol{x})\|_2 \|\boldsymbol{x}\|_2}, i^* = \arg\max_{i \neq y} f(\boldsymbol{x})_i \tag{43}$$

and

$$\tilde{\boldsymbol{x}} = \arg\max_{\tilde{\boldsymbol{x}} \in \mathcal{B}_\varepsilon(\boldsymbol{x})} \log\left(1 + \exp(-(d(g^*(\boldsymbol{x}), \boldsymbol{x}) - d(g^y(\boldsymbol{x}), x)))\right) \tag{44}$$

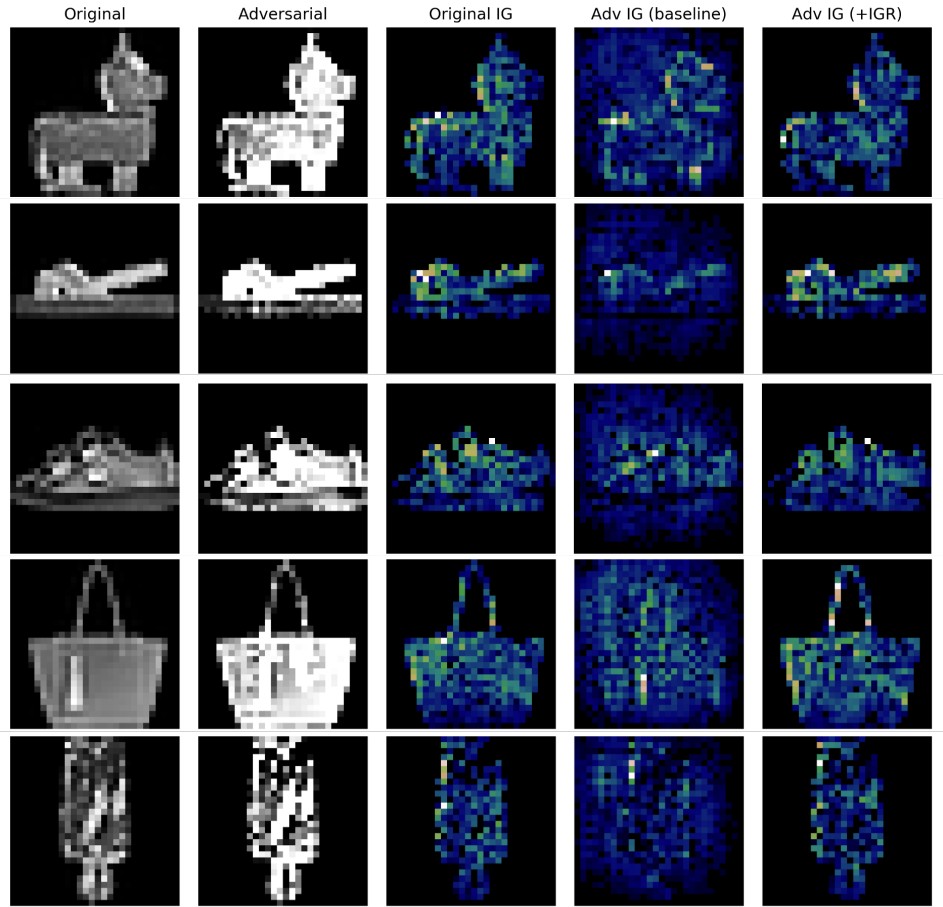

| Original | Adversarial | Original IG | Adv IG (baseline) | Adv IG (+IGR) |

Figure 5: Additional visualization on Fashion-MNIST.

**SSR [33]**

$$\mathbb{E}_{\mathcal{D}} = \left[ \mathcal{L}(\boldsymbol{x}, y; \boldsymbol{\theta}) + \lambda s \max_i \xi_i \right]. \tag{45}$$

$\max_i \xi_i$ is the largest eigenvalue of Hessian matrix $\tilde{\boldsymbol{H}}_{\boldsymbol{x}} = W(diag(\boldsymbol{p}) - \boldsymbol{p}^\top \boldsymbol{p})W^\top$, where $W$ is the Jacobian matrix of the logits vector and $\boldsymbol{p}$ is the probits of the model.

### C.3 Additional visualization of attribution robustness

In this section, additional visualizations are provided in Fig. 5 and Fig. 6 to demonstrate that IGR improves attribution robustness. The original and adversarial images from different datasets are shown in the first two columns. The remaining three columns are IG of the original images on baseline model, IG of the adversarial images on baseline model and IG of the adversarial images on baseline+IGR model, respectively. The baseline model in the visualizations is MART.

The first two columns are the original and adversarial images from Fashion-MNIST. The third column is the IG of the original image. The last two columns are IG of adversarial examples on a baseline model and the baseline model trained with IGR. Both baseline and baseline+IGR models make the correct classifications, while only baseline+IGR protects the model interpretations.

### C.4 Visualization of Kendall's rank correlation and Pearson's correlation

Pearson's correlation against Kendall's rank correlation has been plotted in Fig. 7 under the same setting as Fig. 2a. For the same set of simulations, the corresponding Pearson's correlations are more randomly distributed.

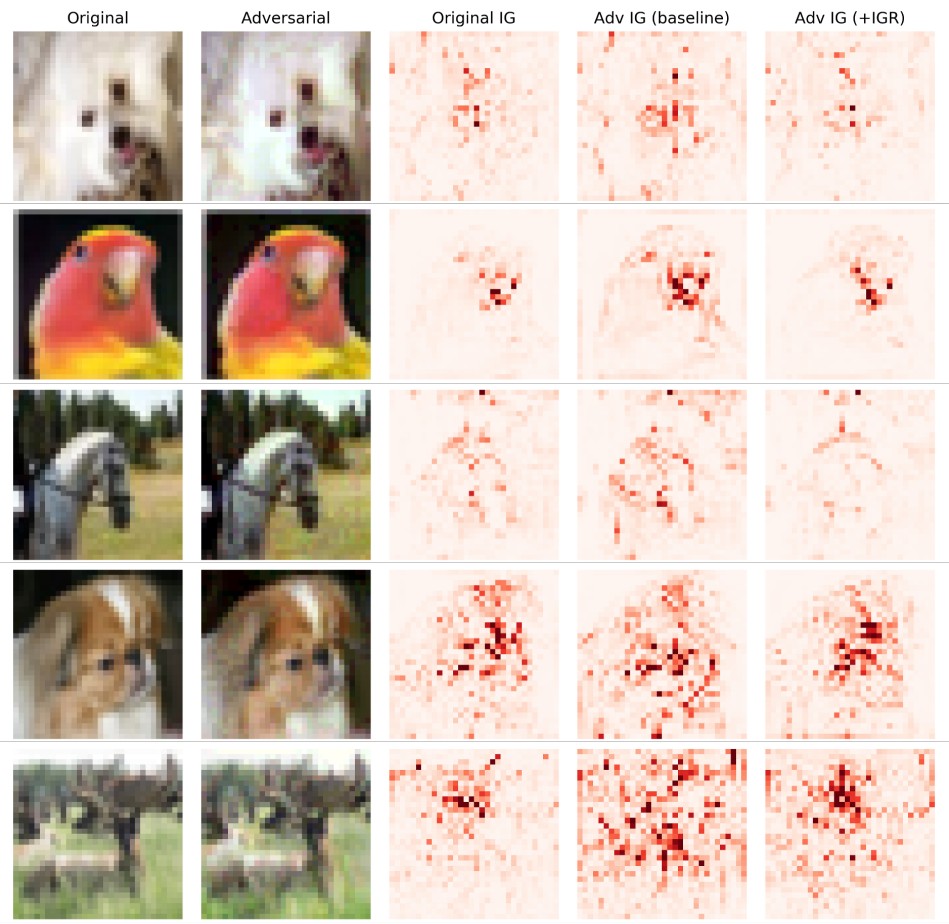

Figure 6: Additional visualization on CIFAR-10.

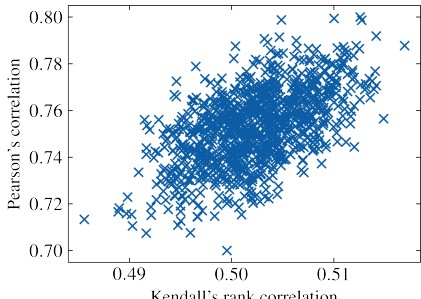

Figure 7: Visualization of Kendall's rank correlation and Pearson's correlation using simulated data.