# OpenReview forum: "Exploiting the Relationship Between Kendall's Rank Correlation and Cosine Similarity for Attribution Protection"
_NeurIPS.cc/2022/Conference — NeurIPS 2022 Accept_

### Official Review · Reviewer_zTbg · 2022-07-08

**Rating:** 7
**Confidence:** 3
**Soundness:** 4 excellent
**Presentation:** 3 good
**Contribution:** 3 good

**Summary:**

This paper theoretically demonstrates that Kendall’s rank correlation of two vectors is correlated to their cosine similarity. Then the authors propose an integrated gradients regularizer (IGR) to regularize the adversarial training methods and make them more robust.

**Questions:**

Could the cosine similarity between integrated gradients be used to select harder adversarial examples during training?

**Limitations:**

No limitation is submitted.

**Strengths And Weaknesses:**

Strengths:
1. The authors demonstrate the relationship of two correlation metrics Kendall’s rank correlation and cosine similarity theoretically and intuitively.
2. This paper is well-written and well-organized.
3. The integrated gradient regularizer helps adversarial training achieve more improvements.
4. The integrated gradients regularizer can make the model more robust.

Weaknesses:
1. Some experimental settings should be described more clearly, such as the top-k metric.
2. No limitations are claimed in the paper.

---

> ### Author Response · Authors · 2022-08-02
> **Response to Reviewer zTbg**
>
> We appreciate your valuable comments. We would like to address the questions in the reviews.
>
> >**Q1**. Some experimental settings should be described more clearly, such as the top-k metric.
>
> The top-k intersection counts the proportion of pixels that coincide in the k most important features ($k$ largest absolute values of the attribution). We also carefully examined the experimental section and clarified some unclear parts.
>
> >**Q2**. No limitations are claimed in the paper.
>
> The main limitation of the proposed method is the excessive memory usage during the training process. As other attribution robustness methods, the most memory usage is from optimization over loss functions containing IG, especially when the number of steps to calculate IG is large. The future work can be focused on reusing the gradients computed during gradient descents and IG computations to reduce memory usage.
>
> >**Q3**. Could the cosine similarity between integrated gradients be used to select harder adversarial examples during training?
>
> As discussed in Section 5.1, we choose to directly use the adversarial examples generated from the corresponding baselines to avoid extra computations. For example, AT+IGR uses the adversarial examples from AT and TRADES+IGR uses adversarial examples from TRADES. Using cosine similarity between integrated gradients to generate perturbed examples would significantly increase the training time, and consume even more GPU memories since multi-step PGD is implemented on IG during each iteration. Moreover, we also believe that using the perturbed samples generated from cosine similarity also decreases the adversarial accuracy.

---

### Official Review · Reviewer_3p2D · 2022-07-11

**Rating:** 7
**Confidence:** 3
**Soundness:** 3 good
**Presentation:** 4 excellent
**Contribution:** 3 good

**Summary:**

### Summary

The authors study an important problem in the space of feature attribution robustness. Based on the cited works in the paper, feature attribution methods are vulnerable to adversarial attacks, such that is possible to obtain remarkably  different attribution by adding an adversarial noise to the input without changing the output label of the model (in the case of classification). This begs the question if it is possible to protect the model against such attribution attacks.  The paper highlights some of the weaknesses in existing attribution robustness (attribution defense) methods such as l_p distance regularizers and Pearson correlation. The authors make a key observation about the Kendall's rank correlation metric which is a commonly used metric for evaluation feature attribution robustness and propose a differentiable alternative based on cosine similarities which can be embedded during training.

The main contributions of the are as follows.
- Theoretically showing the relationship between Kendall's rank correlation between natural and perturbed attribution and their cosine similarity (under certain assumptions).
- Providing a geometric perspective to explain connection between adversarial robustness and attribution robustness
- Proposing a novel Integrated Gradients Regularizer method which (i) Improves attribution robustness and (ii) increases activation consistency for natural and perturbed examples.

Overall it is a good paper and I recommend the paper for an accept.




**Questions:**

### Questions

- [ ] I have not been able to read the prior work in full detail. Can you please clarify if both the prior work https://arxiv.org/pdf/2010.07393.pdf  and the current work seem to involve an outer minimization and an inner maximization to obtain the adversarial perturbed example ? Is this prior work differentiable ?
- [ ] Can you include AdvAAT loss function in Table 1 and highlight the difference between AT and AdvAAT. I think I understand the difference based on going through prior work but would be good to confirm.


**Strengths And Weaknesses:**

### Strengths and Weaknesses


Strengths:
- [ ] Providing a theoretical justification for using cosine similarity and exploiting its relationship with kendall rank correlation metric which seems to not have been done in the prior work in https://arxiv.org/pdf/2010.07393.pdf .
- [ ] Theoretical proof in the Appendix around instability of pearson correlation for small variance.
- [ ] The geometrical perspective with associated figure and explanation is convincing and intuitively explains the claims.

Weakness:
- [ ] One major concern I have with the paper is that while theoretically it is shown the Pearson correlation is unstable and the IGR (Integrated Gradient Regularization) method seems to give best performance, it would also be helpful to compare the it's correlation with Kendall rank correlation. i.e. a figure similar to Figure 2(a) but for the pearson correlation term used in prior work.
- [ ] Suggested though not important for review decision: Some discussion around how the KL divergence term in TRADES might be interacting with IGR to give large gains.

nit:
- [ ] Line 103: Typo paris -> pairs
- [ ] Suggest to cite the following in Related Works
	- [ ] https://arxiv.org/pdf/2206.03178.pdf
	- [ ] https://www.aaai.org/AAAI22Papers/AAAI-3836.HuaiM.pdf
	- [ ]

---

> ### Author Response · Authors · 2022-08-02
> **Response to Reviewer 3p2D**
>
> We appreciate your comments. We would like to address the following questions in the review.
>
> >**Q1**. One major concern I have with the paper is that while theoretically it is shown the Pearson correlation is unstable and the IGR (Integrated Gradient Regularization) method seems to give best performance, it would also be helpful to compare the it's correlation with Kendall rank correlation. i.e. a figure similar to Figure 2(a) but for the pearson correlation term used in prior work.
>
> Pearson's correlation against Kendall's rank correlation has been plotted under the same setting as Figure 2(a). We added one more figure to the Appendix showing the relationship between Pearson's correlation and Kendall's rank correlation. For the same set of simulations, the corresponding Pearson's correlations are more randomly distributed.
>
> >**Q2**. Suggested though not important for review decision: Some discussion around how the KL divergence term in TRADES might be interacting with IGR to give large gains.
>
> We appreciate the suggestion. We also believe that it is worth investigating the reason that TRADES+IGR outperforms other methods. But we also think the discussion of it is outside the scope of this work and would like to consider it as a future work.
>
> >**Q3**. I have not been able to read the prior work in full detail. Can you please clarify if both the prior work [https://arxiv.org/pdf/2010.07393.pdf](https://arxiv.org/pdf/2010.07393.pdf) and the current work seem to involve an outer minimization and an inner maximization to obtain the adversarial perturbed example ? Is this prior work differentiable ?
>
> The mentioned work uses Pearson's correlation as a differentiable surrogate measurement of Kendall's rank correlation, and it inner maximizes the loss function to obtain perturbed examples. We have indicated that Pearson's correlation is unstable as an alternative measurement for attribution difference since two slightly different inputs can have drastically different correlations (see Appendix B). Moreover, since the inner maximization of the prior work requires multi-step gradient descent over Pearson's correlation between integrated gradients, the computation costs are significantly high due to the PGD on integrated gradients. Instead, our work directly uses the adversarial examples from baseline methods (*e.g.*, AT, TRADES and MART) to reduce the computation loads. Besides, our method is also able to maintain the adversarial accuracy, while, according to the results in their paper [1], their method will harm the adversarial accuracy.
>
> [1] Ivankay, et al. "FAR: A general framework for attributional robustness." _arXiv preprint arXiv:2010.07393_ (2020).
>
> >**Q4**. Can you include AdvAAT loss function in Table 1 and highlight the difference between AT and AdvAAT. I think I understand the difference based on going through prior work but would be good to confirm.
>
> The purpose of Table 1 is to provide the loss functions of methods which IGR is embedded to. AdvAAT belongs to one of the other attribution protection methods that we are comparing, and all the loss functions of those comparing methods are listed in Appendix C. We have modified the experiment details to highlight the differences among different methods as suggested. The loss functions of AT and AdvAAT are as follows, where CE refers to cross entropy loss and $\text{PCL} = 1 - (\text{PCC}+1)/2$ is derived from Pearson's correlation (PCC).
>
> |        |                                                                 |
> | ------ | --------------------------------------------------------------- |
> | AT     | $\mathbb{E}[\max_{x'}CE(x', y)]$                                |
> | AdvAAT | $\mathbb{E}[\max_{x'}\{CE(x', y)+\lambda PCL(IG(x), IG(x'))\}]$ |
>
> AT minimizes the cross entropy loss for the worst-case perturbed images which maximize the cross entropy to increase adversarial accuracy. Upon AT, AdvAAT adds a regularizer monitoring the attributions to the maximization problem. It generates perturbed samples that maximize both the cross entropy and the regularizer.
>
> >**Q5**. Typo and related works.
>
> Thanks for pointing out the issues. We have updated the paper to correct the typos and discuss the mentioned works.

---

> > ### Comment · Reviewer_3p2D · 2022-08-08
> > **Thanks**
> >
> > Thanks for addressing the comments !

---

### Official Review · Reviewer_H33g · 2022-07-24

**Rating:** 7
**Confidence:** 3
**Soundness:** 3 good
**Presentation:** 3 good
**Contribution:** 3 good

**Summary:**

This paper proposes integrated gradient regularizer (IGR) for attribution protection. Incorporating IGR is to combine a standard (adversarial) loss function with a similarity function measuring the difference between the natural and perturbed attributions. The authors propose to use the cosine similarity of the integrated gradients (IG) as the similarity function, and they theoretically show that it is an appropriate differentiable replacement for Kendall’s rank correlation, which is a metric quantifying the differences between attributions. The authors also show the downsides of previous attribution protection methods based on $l_p$-norm, along with the shortcomings of standard adversarial training for attribution protection and cosine similarity for adversarial protection. Experiments show that IGR can both contribute to attribution protection and adversarial protection, surpassing previous baselines.

**Questions:**

In the caption of Figure 2, the authors say that “Solid ball represents the untrained attribution surface $g$ and the dashed ball is the trained surface $g^*$.” What do “trained” and “untrained” mean?

**Limitations:**

I suggest the authors describe the concept of attribution protection in more detail in the preliminaries, and emphasize its significance, for the concept seems to be not very popular in the ML community.
I suggest the authors make a box plot of +IGR and baseline models to show the variance on multiple samples.

**Strengths And Weaknesses:**

Strengths:

The proposed IGR is simple and effective, strengthening both adversarial and attribution robustness. The advantages are shown both theoretically and empirically. The presentation is generally clear.

Weaknesses (and suggestions):

- The title is too narrow. I suggest not to mention the specific metric “Kendall’s Rank Correlation”.
- The detail of choosing $\lambda$ is missing. What value is set for it and how does it affect the results?
- In section 4, I suggest the authors first point out the implications of (a) the angle between the attribution vectors and (b) their magnitudes, and state the importance of minimizing (a) as well as maximizing (b). When explaining the drawbacks, I suggest the authors focus on the characteristics of the methods for minimizing (a) and maximizing (b).

---

> ### Author Response · Authors · 2022-08-02
> **Response to Reviewer H33g**
>
> We appreciate your comments. We would like to address the following questions in the review.
>
> >**Q1**. The title is too narrow. I suggest not to mention the specific metric “Kendall’s Rank Correlation”.
>
> This work connects Kendall's rank correlation and cosine similarity in a qualitative way, which is then used to improve the attribution robustness. We would like to highlight this relationship as one of the major contributions of this work in the title.
>
> >**Q2**. The detail of choosing $\lambda$ is missing. What value is set for it and how does it affect the results?
>
> We investigated the impact of $\lambda$ by experiments on different choices of $\lambda$ values. As mentioned in Section 5, $\lambda$ is the hyperparameter to balance the classification accuracy and attribution robustness. For example, the following tables show the results of MNIST and Fashion-MNIST using different values of $\lambda$ on AT+IGR. The numbers are evaluated from a validation set with 10,000 images. We can observe from the results that $\lambda$ with higher Kendall's rank correlation usually corresponds to lower classification accuracy and vice versa. In the paper, we reported the evaluation of the test set using models with the best Kendall's rank correlation (the corresponding $\lambda$ is marked with * in the table).
>
> - MNIST
>
> | $\lambda$ | Natural | FGSM   | PGD20  | CW     | top-k  | Kendall |
> | --------- | ------- | ------ | ------ | ------ | ------ | ------- |
> | 2.0       | 0.9952  | 0.9933 | 0.9925 | 0.9922 | 29.85% | 0.1165  |
> | 3.0*       | 0.9941  | 0.9930 | 0.9918 | 0.9918 | 34.07% | 0.1457  |
> | 4.0       | 0.9947  | 0.9936 | 0.9917 | 0.9916 | 35.02% | 0.1383  |
> | 5.0       | 0.9951  | 0.9941 | 0.9934 | 0.9930 | 30.79% | 0.1357  |
>
> - Fashion-MNIST
>
> | $\lambda$ | Natural | FGSM   | PGD20  | CW     | top-k  | Kendall |
> | --------- | ------- | ------ | ------ | ------ | ------ | ------- |
> | 2.0       | 0.8378  | 0.7596 | 0.7207 | 0.7159 | 50.71% | 0.3219  |
> | 3.0       | 0.8015  | 0.7831 | 0.7616 | 0.7557 | 51.53% | 0.3310  |
> | 4.0*      | 0.8021  | 0.7482 | 0.7203 | 0.7164 | 56.17% | 0.3714  |
> | 5.0       | 0.8217  | 0.7596 | 0.7331 | 0.7281 | 52.57% | 0.3391  |
>
>
> >**Q3**. In section 4, I suggest the authors first point out the implications of (a) the angle between the attribution vectors and (b) their magnitudes, and state the importance of minimizing (a) as well as maximizing (b). When explaining the drawbacks, I suggest the authors focus on the characteristics of the methods for minimizing (a) and maximizing (b).
>
> Thanks for the advice. We have revised the related section in the paper based on the suggestion.

---

> > ### Comment · Reviewer_H33g · 2022-08-08
> > **Response to Authors**
> >
> > Thanks for your reply!
> >
> > The revised version is clearer to me. Why not put the tables of choices of $\lambda$ in the appendix?

---

> > > ### Author Response · Authors · 2022-08-08
> > > **Response to Reviewer H33g**
> > >
> > > Thank you very much for your reply. We will follow your suggestions and include the tables and discussions on choosing $\lambda$ in the final version.

---

### Author Response · Authors · 2022-08-02
**Changes in the revised paper**

We would like to thank all the reviewers for your valuable comments. We have revised the original paper based on the comments and suggestions. The following lists the major changes to the paper, and all the parts have been highlighted in the paper using blue texts. We hope the revision could address your questions.

1. Reorganized part of the contents in Section 4.
2. Added an extra figure showing the unstable Pearson's correlation in Appendix C.4.
3. Modified the implementation details in Appendix C.2 to highlight the baselines differences.
4. Corrected typos.

---

### Meta-Review · Area_Chair_EjLv · 2022-08-25

**Recommendation:** Accept
**Confidence:** Less certain

**Metareview:**

Reviewers all find this paper presenting both good theoretical findings and empirical results for an important problem (feature attribution robustness). The approaches authors used in connecting the relationship between Kendall’s rank correlation and cosine similarity, as well as the geometric perspectives, are well received. The presentations are well written, with minor places for quick improvements.

Reviewers have raised various weakness points but we agreed most of them are minor, do not affect the contribution of this paper, and/or can be fixed without much effort.

Overall we recommend acceptance and would like to encourage the authors to further improve this paper presentation following reviewers’ suggestions in the next version.

**Award:**

No

---

### Decision · Program_Chairs · 2022-09-14

Accept